# Urinary Tract Virome as an Urgent Target for Metagenomics

**DOI:** 10.3390/life11111264

**Published:** 2021-11-19

**Authors:** Agata Salabura, Aleksander Łuniewski, Maria Kucharska, Denis Myszak, Barbara Dołęgowska, Kazimierz Ciechanowski, Karolina Kędzierska-Kapuza, Bartosz Wojciuk

**Affiliations:** 1Clinic of Nephrology, Internal Medicine and Transplantation, Pomeranian Medical University in Szczecin, 70-123 Szczecin, Poland; kazciech@pum.edu.pl; 2Department of Immunological Diagnostics, Pomeranian Medical University in Szczecin, 70-123 Szczecin, Poland; aleksander.luniewski@gmail.com (A.Ł.); kucharskam01@gmail.com (M.K.); denis.myszak@gmail.com (D.M.); barbara.dolegowska@pum.edu.pl (B.D.); bartosz.wojciuk@pum.edu.pl (B.W.); 3Center of Postgraduate Medical Education in Warsaw, Department of Gastroenterological Surgery and Transplantology, 137 Wołoska St., 02-507 Warsaw, Poland; karolina.kedzierska@interia.pl

**Keywords:** urobiome, virome, phageome, new sequencing methods

## Abstract

Virome—a part of a microbiome—is a term used to describe all viruses found in the specific organism or system. Recently, as new technologies emerged, it has been confirmed that kidneys and the lower urinary tract are colonized not only by the previously described viruses, but also completely novel species. Viruses can be both pathogenic and protective, as they often carry important virulence factors, while at the same time represent anti-inflammatory functions. This paper aims to show and compare the viral species detected in various, specific clinical conditions. Because of the unique characteristics of viruses, new sequencing techniques and databases had to be developed to conduct research on the urinary virome. The dynamic development of research on the human microbiome suggests that the detailed studies on the urinary system virome will provide answers to many questions about the risk factors for civilization, cancer, and autoimmune diseases.

## 1. Introduction

It is now known that various microorganisms play an important role in maintaining homeostasis across many human systems. The disturbance in what was described as “microbial flora” often leads to various immunity-related complications not only in the gastrointestinal tract, but also the skin, urinary tract, or even central nervous system. With recent discoveries showing that viruses are in fact present in almost every biome of the human body, it may be assumed that disturbances in the human virome may play a similar role to that of bacteria in the pathology of many known diseases.

To date, viruses have been known to occupy various niches, not only within the human body, but also within cells themselves, with the most prevalent viral group being the bacteriophages [1]. Some viruses survive as an integrated part of the human genome and are known as human endogenous retroviruses (HERVs), which, in certain cases, is beneficial to their host, but can also result in tumor transformation and autoimmune diseases [2]. Meanwhile, bacteriophages are considered to be a group of viruses that are responsible for keeping bacterial numbers in check through lysis. However, at the same time, they are known to transmit key virulence factors, such as exotoxin production, adhesion, and invasion and evasion from immunity [3]. On the other side, phages are considered as a possible alternative to antibiotics, and represent certain anti-inflammatory characteristics [4,5].

Urine and the urinary tract have been considered sterile for a long time and the detection of any microorganism within was considered a clear sign of pathology. Only recently has that dogma started to change, as an increasing number of discoveries proved urine to be in fact colonized by various microorganisms including viruses. It was mostly because of the development of new, culture-independent research techniques, such as next generation sequencing (NGS), which made it possible to discover different new components of the human microbiome, previously either undiscovered or not considered as a constant part of it [6].

Urinary tract infections are mainly caused by bacteria. Viruses are recognized as a rare etiological agent in urinary tract infections. The most common are adenoviruses, mainly in childhood. However, the epidemiology appears significantly different in people with decreased immunity related to the following: immunosuppressive treatment after organ transplantation, oncological treatment, immunodeficiencies including AIDS, and newborns. Recognition of etiology and learning about new viruses is possible thanks to NGS and its development.

Below, we will present and discuss the most relevant discoveries on urinary virome in health and pathology as well as the specificity of virome-directed metagenomics. Phageome and SARS-CoV-2 will be considered separately. The most clear data refer to the group on immunosuppressive therapy. However, the research on other groups is ongoing as these require larger study groups and better profiling.

## 2. Eukaryotic Virome in Health

Human papillomaviruses (HPV), BK virus (BKV), JC virus (JCV), and Torque teno virus (TTV) are the main eukaryotic viruses that create what is known as a urinary virome. While human papillomaviruses mostly have double-stranded DNA and are known for their genitourinary tract tropism, BK and JC viruses included in the family of Polyomaviridae are also double-stranded DNA and are associated with diseases such as nephropathy in renal transplant (BK) virus and progressive multifocal leukoencephalopathy (JC) virus. Meanwhile, TTS has single-stranded circular DNA and is not commonly associated with any disease. Some believe that perfectly adapted viruses should not impact its hosts in any way, and as a result they would be clinically silent. The knowledge about those viruses was very limited before the next generation sequencing era [7]. Before, the detection of viral presence relied mostly on the presence or absence of clinical symptoms such as diarrhea, increased aggressiveness leading to biting, or cough, which also led to higher transmission of viral particles. Despite this, the newly discovered viruses that can be found in the urine of healthy individuals indicate that being ‘silent’ allows them to create an ideal equilibrium with the host, and even in some cases provide certain advantages to the infected cells [8]. JC is an example of a common virus with unique properties that has been detected among healthy individuals. In the general population of the USA and Brazil, approximately 30% of individuals display evidence of active replication of this virus in urine. It has been proposed that this virus represents a protective potential in chronic kidney disease [9].

## 3. Virome during Immunosuppressive Treatment

A variety of nephrourinary conditions coexist in immunocompromised hosts like solid organs recipients [10]. Immunosuppressive drugs are known to influence cell-mediated immunity, which can have detrimental effects such as reactivation of hepatitis C virus in liver transplant recipients. Most important viruses responsible for viral nephropathy following immunosuppressive treatment are Cytomegalovirus, Herpes Simplex virus, Varicella Zoster, Epstein–Barr virus, Human Herpes virus 7 and 8, and BK/JC polyomaviruses [11]. Concomitant reactivation of any of these influences the urinary tract virome as a whole.

In a cohort study comparing kidney transplant recipients with healthy controls, it was found that the kidney recipients possessed many unique viruses such as Junin virus, Cotesia congregate, or Pseudocowpox virus, and these had not been recognized as a component of human virome previously. Depending on a kidney graft status, different proteins corresponding to specific viruses were detected. Patients with stable graft status had a higher prevalence of Junin virus proteins when compared with healthy controls and the other groups. More interestingly, evident conditions such as BK virus reactivation produced different urinary viral profiles. Stable kidney recipients expressed overall less viral peptides when compared with kidneys either with chronic or acute rejection as well as BK nephropathy. In transplant recipients undergoing BK virus infection, unique viral peptides were also found [12]. These peptides belonged to viruses absent in other groups such as Trichoplusia ni ascovirus. In another study focusing on the comparison between BKV+ and BKV− kidney graft recipients, it was found that the BK virus responsible for post-transplant infection was polymorphic in sequences responsible for VP1, VP2, and large T antigen proteins. BK virus positivity also seems to correlate with the absence of certain viruses from the family of single-stranded circular DNA viruses Anelloviridae, which is probably a result of competition for intra-nuclear replication sites between these viruses [13]. Overall, it was proven that significant differences exist in urinary virome depending on the transplanted kidney status.

Another example of viral competition is infection with either BK or JC polyomavirus. Both of these are responsible for post-transplant viral nephropathy. JC virus, which also causes progressive multifocal leukoencephalopathy, is associated with far fewer complications during kidney infection, while BK virus infection may result in graft loss [14]. According to one study, the presence of viremia from JC polyomavirus made the infection with BK virus less likely, and vice versa. Additionally, JC polyomavirus has been associated with lower acute rejection rate and improved graft survival when JC viruria occurred, probably owing to the lower chance of BK polyomavirus co-infection [15]. JC virus is also associated with a lower risk of non-diabetic nephropathy by interacting with apolipoprotein gene APOLI1. Certain variants of this gene are associated with kidney disease [16].

## 4. Virome in Oncological Patients

It has been well recognized that viral infections are responsible for 12–15% of human cancers [17]. Among oncogenic viruses, Epstein–Barr virus (EBV), human T lymphotropic virus type 1 (HTLV-1), hepatitis B virus (HBV), human papillomavirus (HPV), hepatitis C virus (HCV), Kaposi’s sarcoma-associated herpesvirus (KSHV, also known as human herpesvirus 8 or HHV8), and Merkel cell polyomavirus (MCV or MCPyV) [18] are worth mentioning. Moreover, endogenous viruses have been recognized to be possibly responsible for carcinogenesis such as well implemented in human genome endogenous retroviruses (ERVs), which in some instances were reported to promote tumorigenesis [19]. As of late, the appearance of new tools such as The Cancer Genome Atlas has made it possible for researchers to investigate causality between viral infections and tumor cell neoplasia and, thanks to this, the relationship between carcinogenesis and urine virome has already been a subject of a several research projects [20,21,22].

Recently, it has been found that there is a possible connection between oncogenic human viruses and the occurrence of bladder cancer. In the PicAxe platform, DNA sequences from cancerous and non-cancerous tissues were analysed, and it was found that 19/268 tumor samples contained DNA belonging to the HPV virus, while 1 sample had an integration from the BKV virus [21]. Another platform—VIcaller—identified clonal integration of genetic material derived by HPV and BKV viruses with the DNA derived from tumors of suspected viral etiology. With the help of this software, the researchers were able to analyse high-throughput sequencing reads from numerous tumours of human origin, and confirmed the presence of viral DNA in four tumour samples with confirmed viral etiology by finding HPV and BKV viral integrations [20]. In the future, genome-based tools may provide a useful tool to screen for early-stage cancers caused by an already integrated virus. One study that implemented the classical PCR method found an HPV prevalence rate of 63.9% in urine samples obtained from HIV positive patients, compared with 70.1% in cervical samples. The finding of HPV genetic material in urine (with HPV-16 being the most common viral type) was also often associated with cytological findings such as atypical squamous cells of undetermined significance and squamous intraepithelial lesions. Similar studies have yet to be performed for BKV viruses, as these may also be responsible for urinary tract neoplasms [23].

The bladder tumorigenesis appears unclear at this moment. However, aforementioned discoveries and possible future development of bioinformatic tools will bring the recognition of underlying mechanisms closer. All these may also create a need for the development of new screening methods based on detecting viral genetic material in urine samples.

## 5. Phageome

Bacteriophages are by far the most prevalent group of human virome and they help to shape the human microbiome across many systems and niches [24]. For example, Manrique et al. created a concept of a healthy gut phageome—that is, a virome consisting of phages essential for maintaining homeostasis in the gastrointestinal system. According to this study, a total number of 23 phages were found in a group consisting of healthy individuals. The same group of phages was present to a far lesser extent in a group of patients with existing gastrointestinal diseases such as ulcerative colitis or Crohn’s disease. This led the researchers to conclude that a healthy gut phageome provides a fundamental input to preserve the function of the gut microbiome and is essential for the gastrointestinal system to function correctly [24]. Recently, the discovery of human fecal matter abundant crAssphage [25] as well as the creation of gut phage databases further helped to expand the gut phageome research [26]. Other important phageomes that were studied include the following: skin phageome [27], respiratory phageome [28,29], and oral cavity phageome [30].

As phages are the amplest group of the human virome, it may be assumed that their presence is important for preserving homeostasis of most systems [1]. One study confirmed the presence of phages in urine of 46.1% of clinical samples. In this study, viruses showed morphological traits of Siphoviridae and Podoviridae phages and were directed against *Escherichia coli* and *Pseudomonas aeruginosa*. The authors suggest that the presence of phages may interfere with the quantification of bacteria in biological samples [31]. Meanwhile, Garretto et al. found an abundance of phage protein-coding sequences in the urine of healthy and overactive bladder women using tool virMine sequences directed against bacteria such as Streptococcus agalactiae or Lactobacillus helveticus [32]. Another study detected a high number of viruses with the majority (>99%) of sequences belonging to phages such as Lambda phage, Staphylococcus phage PH15, and E. coli phage phiV10 [33].

When it comes to phage life cycle, it has been reported that phages isolated from the genitourinary tract possessed a high prevalence of integrase genes, which suggests a mostly lysogenic life cycle [34] This appears analogous to the status in the gastro-intestinal system, where it is considered that every bacterial cell has at least one prophage within. At the same time, the concentration of bacteriophages was found to be low (101–102 pfu/mL), which may be the effect of the lower durability of the phages in urine [35].

The role of bacteriophages in many other aspects of the urinary tract remains unclear and warrants further investigation [36]. While antibiotic resistance becomes an even greater challenge with every passing year, phage therapy provides an antimicrobial alternative, and hence a promising option for treating urinary tract infections. As a viable treatment option for UTIs, phages are being examined not only for their lytic enzymes and proteins, which could help fight the bacteria, but also as complete functioning phages themselves. This is expected to boost already existing antibiotic therapies in the form of ‘phage cocktails’ and genetically engineered bacteriophages [37,38]. One report points to five phages from Podoviridae, including some that possess biofilm-reducing capabilities, as possible candidates for Proteus mirabilis urinary tract infections therapy [39]. Another study reports the efficacy. There are also other reports on phages that may help reduce the biofilm production of other bacteria such as Pseudomonas aeruginosa [40], Staphylococcus epidermidis [41], and Escherichia coli [42].

## 6. Virome Researching Methods

The shift in metagenomic approach and appearance of new generation sequencing (NGS) helped change research on the human microbiome. Older techniques that relied on cultures and light microscopy made these investigations much more challenging. The appearance of ribosomal 16S coding-regions-based sequencing made it possible to discover microorganisms living in sterile-considered environments such as the urinary tract [6]. As NGS does not require large amounts of DNA in a sample, the discovery of many new organisms in the human microbiome has been accelerated.

Concerningly, this type of sequencing comes with its own limitations when it comes to studying viruses, as 16S ribosomal RNA sequence is only found in bacteria. Because of this, in order to implement metagenomics into viromic studies, certain modifications had to be made. The first step to obtaine a viable viral genome is sample purification. This is important because it determines the amount of background ‘noise’, mostly consisting of bacterial and eukaryotic DNA. Differences in available sampling techniques may skew the results towards one group of microorganisms, which may be responsible for drastic differences between studies investigating the same part of the virome. As a result, various techniques are proposed to enhance the content of viral DNA such as CsCl gradient ultracentrifugation, special filters that purify the sample of bacterial DNA, epifluorescence microscopy, automated extraction platforms used in combination with PCR, and flow-cytometry-based methods. All these methods are meant to purify and maximize the highest possible level of virus like particles (VLPs) and are summarized briefly in Table 1 [43,44].

Biological samples that contain viral genetic material are subsequently sequenced. As 16s-based sequencing is not available to study viruses, methods such as shotgun sequencing or protein-based virus recognition are used. This allows to bypass the need for having a specific sequence, but also comes with certain issues such as high-background noise.

When it comes to shotgun sequencing, its name is meant to represent the random, shotgun-like break up of long DNA sequences that are later sequenced in the form of reads, which are subsequently put together by bioinformatic tools into larger, continuous sequences [45]. Once various gene sequences are obtained, these are further analyzed using software that integrates all the obtained data in the context of the whole possible genome of the microorganism, and eventually-functional proteins. The independence of shotgun sequencing from 16S rRNA-based sequencing means that it can be often used in clinical settings, when the clinical samples are not always suitable for culture-dependent methods and, furthermore, it would be impossible to detect the scarce material from viruses without knowing the specific primers.

The platforms used in next generation sequencing differ, with protocols used to address the read length, time, raw error rate, and cost [46]. Those differences are highlighted in Table 2.

After obtaining viral sequences, bioinformatic analysis is required. For proteins-coding sequences, once open reading frames are determined, the proteins of a specific virus may be compared with amino acid sequences in available databases. Most of these consist mostly of cellular microorganisms’ sequences, but there are also some dedicated to viral sequences such as CheckV [48] or VIBRANT [49]. However, it may be necessary to employ other methods for viruses that were not previously identified. These include machine learning such as clustering viral sequences by composition [43]. Lysogenic viruses constitute a significant challenge as these may stay dormant within the host’s cell. Nowadays, researchers have access to numerous tools to complete the process.

Recently, the appearance of tools such as virMine, which can ‘scan’ huge datasets in search of viral sequences, enables the discovery of viruses hidden within very complex metagenomic samples [50]. Other tools, such as VIcaller, are able to detect viral sequences hidden within the eukaryotic genome [20].

As mentioned, another challenge comes when searching for new organisms. The traditional PCR-based approach proves insufficient, as it only allows the detection of new organisms using degenerated primers. Hence, its usefulness is limited to a group of more or less related organisms. Because of this, strategies such as sequence independent amplification, DNA microarrays, and NGS proved extremely efficient in finding novel infectious agents [51].

## 7. SARS-CoV-2

Even though it is not a part of the physiologic human virome, SARS-CoV-2 is reported to have an effect on other systems aside from the respiratory tract itself. As a result, numerous diagnostic tools that were developed to detect viruses colonising different human organs can become useful in diagnosing COVID-19-related disease as well. For instance, the angiotensin converting enzyme 2 receptor (ACE2) for the spike protein responsible for the entry of COVID-19 into the host cell is expressed in number of tissues (lungs, oesophagus, ileum, colon, kidney, myocardium, bladder, and oral mucosa), and it is likely that the virus may be responsible for a number of changes in these organs [52]. It was also theorized that, because of the presence of ACE2 receptors in CNS, the virus may be responsible for some cases of hyposmia and acute respiratory failure in COVID-19 patients [53].

As ACE2 receptor is expressed in urinary tract organs, there are reports of SARS-CoV-2 being isolated from urine. One patient undergoing COVID-19 infection was hospitalized in Guangzhou, China and, upon testing, the patient’s urine samples were positive for the viral DNA. Subsequently, it was proven that viral particles isolated from urine were infectious towards Vero E6 susceptible cells in an immunofluorescent assay [54]. As the virus was also isolated from stool samples, these routes may pose an important factor in transmission of the virus [55]. It has been also reported that, because of the presence of SARS-CoV-2-specific amino acids in urine, the urine-directed diagnostic approach can be potentially useful as a screening or an ongoing COVID-19 diagnostics tool [56].

Recently, the presence of SARS nucleocapsid protein (SARS-CoV-2 N) in urine was evaluated in a single center prospective observational study as a possible identification tool of patients at high risk of developing acute kidney injury (AKI) in their course of COVID-19 infection. It was confirmed that, on days 1, 3, and 8 of intensive care unit (ICU) stay, the presence of SARS-CoV-2 N was associated with the AKI stage. This correlation was further improved by combining the SARS-CoV-2 N levels with plasma albumin measurements, which not only allowed for better identification of patients at risk of AKI, but also allowed researchers to predict the length of ICU supportive care and the risk of premature death from COVID-19 infection [57]. Another diagnostic method that could possibly help identify patients at risk of death from COVID-19 was proposed by Caricchio et al. In their study conducted on a group of 513 patients 12 variables were associated with COVID-19 cytokine storm (COVID-CS). After dividing these variables into three clusters, researchers developed criteria for COVID-CS, which helped to identify patients with longer hospitalization (15.1 ± 13 vs. 5.7 ± 6.7) and increased mortality (28.8% vs. 6.6%) [58].

Organotropism of COVID-19 was further analyzed by Puelles et al. conducted in a study that assessed SARS-CoV-2 viral load in tissue samples collected from 22 patients who died from COVID-19. Based on their results, it was reported that the virus expresses a significant renal tropism, which is likely a result of high expression of genes involved in SARS-CoV-2 infection in kidneys such as ACE2, transmembrane serine protease 2 (TMPRSS2), and cathepsin L (CTSL). What is more, in the same study, other tropisms have been identified as high viral-load was detected in other organs such as the pharynx, heart, liver, and brain [59]. Another study that post-mortem analyzed the presence of SARS-CoV-2 viral RNA in kidneys of the patients who had died of COVID-19 confirmed that 60% of individuals presented detectable viral load in kidneys, as did 72% of patients with acute kidney injury diagnosed pre-mortem. The same study also revealed an important association between the reduction in patients’ survival time and SARS-CoV-2 kidney positivity with a hazard ratio of 3.7 [60].

## 8. Discussion

As new bioinformatic tools are being developed, the discovery of previously obscured viruses in many systems becomes possible. This is true not only for previously thought to be sterile urine, but also for other environments such as ascitic fluid or the central nervous system. In light of recent discoveries, it seems very likely that viruses play an important role in the homeostasis of the genito-urinary system. Urine itself is a challenging environment, as the concentrations of viral particles within it are much scarcer. This makes the research particularly more difficult, and is probably the reason for which the urinary microbiome has remained undiscovered longer than other biomes. New scientific methods, which allow finding viral genetic material integrated into the host’s cells, made it recently possible to study hidden viruses within genomes. It is especially important as these viruses, such as HPV, are often responsible for oncogenesis, and detecting them within cancer cells may shed some new light on the pathophysiology of many types of cancers. NGS-based research also benefits from the development of new bioinformatic methods, as metagenomic data obtained often require complex analysis in order to find and recognize sequences of viral origin.

The discovery of a variety of viruses within the urinary tract makes it clear that further studies are warranted in order to establish viruses that represent the norm, and which may be a cause or an effect of a pathology. Comparative studies that examine different viromes in patients with ongoing immunosuppression may provide an exceptionally useful input about this issue, as the reduction in cell-type immunity often leads to the proliferation of opportunistic viruses, which are normally absent in healthy individuals.

On the other hand, bacteriophages have been proven to important part of the human microbiome, not only because of their contribution to pathological processes, e.g., transmitting various virulence factors between bacteria, but also because they may play a part in antibacterial defense. In fact, it was confirmed that lytic bacteriophages can be responsible for false-negative results of urine samples, as the bacteria they affect undergo lysis before urine itself is analyzed.

All of these as well as a recent discovery of infectious SARS-CoV-2 isolated from urine prove that urinary virome is an important element of the human microbiome, which directly corresponds to health and provides easily-accessible information about pathologies and disturbances in homeostasis within our organisms.

However, a more integrative bioinformatic approach is needed in order to uncover the complex interplay between the bacterial microbiome, virome, and host cells. Such an approach should consider both direct and indirect interactions between these. Therefore, a strong effort to optimize different sorts of metagenomic data and other-omics-derived data must be implemented. This is particularly valid for urinary tract disorders as the biofilm formation represents a possible key factor in subsequent pathologies and is also recognized as one of the major challenges in current biomedicine.

## Figures and Tables

**Table 1 life-11-01264-t001:** Characteristics of VLP sampling techniques.

VLP Sampling Technique	Characteristics
CsCl centrifugation	May skew the results towards isolating specific group of phages, while providing very pure sampling
Viral quantification method	Underestimation of VLP in samples
0.2 μm filters	Depletion of large viruses and reduction of recovered viral DNA
Automated extraction platforms	Used in conjunction with qPCR and droplet PCR to achieve high sensitivity

**Table 2 life-11-01264-t002:** Sequencing platform types.

Sequencing Platforms Available	Examples	Methods
Illumina	iSeq, MiSeq, MiniSeq, NextSeq, HiSeq, NovaSeq	Bridge amplification (DNA molecules are attached to a flow cell and amplified locally)
Thermo Fisher Scientific	Ion Torrent platform	DNA cloning on a bead within an emulsion
BGI	BGISEQ	Local DNA cloning on a flow cell producing clonal DNA nanoballs
Oxford Nanopore Technologies	MiniION, GrindION, PromethION	Single-stranded DNA guided through protein nanopores gathering DNA sequences with electrical current [47]

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
