# Peer review of "Urinary Tract Virome as an Urgent Target for Metagenomics"

_life, 2021, doi:10.3390/life11111264_

Round 1

Reviewer 1 Report

Thank you for giving me the opportunity to review this manuscript. It is really intresting and the subject is yet to be explore. Urinary tract virome is far from beeing completly elucidated. The previous reserch conductted by other reserchers brought some inside about this subject. I believe the manuscript is well structured and targeted all important aspects of the subject. I recomened it for publication.

Author Response

Thank you for your opinion. 

Reviewer 2 Report

In this review article, Salabura et al. discuss recent advances in urinary tract virome in the context of diseases, including the ongoing COVID-19 pandemic. The review is timely and interesting. I recommend to discuss the role of SARS-CoV-2 due to importance:

  1. Please include a paragraph discussing SARS-CoV-2 organ tropism, including the lung, heart and kidneys. In addition, the important literature should be cited accordingly (e.g. Puelels et al. NEJM 2020, Braun et al. Lancet 2020)
  2. Moreover, evidence for detection of viral particles in urinary samples should be discussed (e.g. Tampe et al. Frontiers Medicine 2021)
  3. Finally, hyperinflammatory syndromes attributed to SARS-CoV-2 should be mentioned and discussed (e.g. Caricchio et al. ARD 2021)

Author Response

We developed the section about the role of SARS-Cov-2 based on that publications. 

Reviewer 3 Report

The article is consistent within itself. The references are relevant and recent. The cited sources are referenced correctly. Appropriate and key studies are included. The paper is comprehensive, the flow is logical and the data is presented critically.
However, there are some specific comments on weaknesses of the article and what could be improved:

Major points - none
Minor points
1. The tables are not cited within the text.

2. What is the unit of the "unique viral proteins".

3. Section 7 - on SARS-CoV-2 - it should be clarified that this virus is not a part of the human virom/microbiom. additionally, auhtors have to explain why they included it here.

Author Response

We improve the points you mentioned. 

Reviewer 4 Report

The authors present a review of a field which may still be too young for a review. However, the review is well executed none the less. Some passages need an english language check, and some sections need more appropriate references and more accurate description of the state of the art (see below). Due to the lack of line numbers, my suggestions below are provided in the format pageXparagraphY.

p2p4: What is JC and BK virus? The reader needs an intro. Are they DNA or RNA viruses? Double or single-stranded? Are they related to TTV or any other known viral classes?

p2p4: This is an interesting opinion, but an opinion none-the-less, so the text should be formulated as such. Surely there are numerous examples of “well-adapted” viruses that cause illness or kill. Also, numerous chronic diseases exist for which no cause is known yet, but they could be caused by some of these “asymptomatic” infections. HPV is a classic example of an “asymptomatic” virus found later to cause cancer. Examples have also been found recently where people with chronic fatigue syndrome were shown to carry certain anelloviruses, while certain polyomaviruses have been linked to spontaneous abortions as well. So although the "good virus" hypothesis is promising, it might also be that we don't know enough yet.

p3p1: The detection of soybean and moth viruses sounds like false positives and should not be reported in a review like this. False positive matches are rampant in virome literature because researchers have limited knowledge in reference-independent virome analysis techniques.

p3p1: Are BK viruses and the anellovirus related? Do they share the same replication machinery? If yes then this should be stated clearly earlier when BK virus is first mentioned.

Table 1: This table can be misleading as it gives the impression that kidney graft and control status is linked to those particular viruses consistently. I am concerned that most of the virus matches were chance findings from individual studies and do not replicate across different studies. Also most of the virus matches in column 3 look like false positives arising from a failure to employ reference-independent techniques that are crucial for virome analysis. As you know, most viruses in most viromes are novel, and matching them against known viruses in irrelevant databases gives misleading results as seen here, and such false results should not be propagated further in a review such as this. Unless they were reproducible accross several studies, in which case I'd believe them, but then references should be provided.

p3p4: replace “up to date” with “recently”. Also, “it was _found_ that”

p4:p3 The phage section is missing key gut phage results that need to be mentioned in the first paragraph: 1) Crassphage 2) The human gut phage databases recently published by the Lawley and Kyrpides groups.

p5p2: next generation sequencing, not “new” generation sequencing

Table 2 needs references

p5p5: The point about independence of 16S from shotgun making it suitable for clinical settings where culturing is difficult is very unclear. What is mean by this?

p6p2: CheckV and VIBRANT and viralVerify are the best tools currently for identifying viral sequences within virome assemblies. They work by calling ORFs in assembled contigs and comparing them internally against a database of viral proteins (so the users don’t need to call ORFs themselves). These tools do a good job of detecting novel viruses as well. These should be mentioned here instead of VOGdb etc as the latter is outdated and the former are the state of the art. Sequence-composition-based methods such as virfinder, deepvirfinder, etc. do not work well and should not be mentioned. Neither should virMine, as it is obsolete at this stage.

Author Response

Thank you for your extensive review. We tried to improve every point you mentioned and higlighted 
